# Open-Bundle Structure as the Unfolding Intermediate of Cytochrome *c*′ Revealed by Small Angle Neutron Scattering

**DOI:** 10.3390/biom12010095

**Published:** 2022-01-07

**Authors:** Takahide Yamaguchi, Kouhei Akao, Alexandros Koutsioubas, Henrich Frielinghaus, Takamitsu Kohzuma

**Affiliations:** 1Institute of Quantum Beam Science, Graduate School of Science and Engineering, Ibaraki University, 2-1-1 Bunkyo, Mito 310-8512, Ibaraki, Japan; takahide.yamaguchi.qbs@vc.ibaraki.ac.jp (T.Y.); kou.060222@gmail.com (K.A.); 2Frontier Research Center of Applied Atomic Sciences, Ibaraki University, 162-1 Shirakata, Tokai 319-1106, Ibaraki, Japan; 3Jülich Centre for Neutron Science JCNS-4 at Heinz Maier-Leibnitz Zentrum, Forschungszentrum Jülich GmbH, Lichtenbergstrasse 1, D-85747 Garching, Germany; a.koutsioumpas@fz-juelich.de (A.K.); h.frielinghaus@fz-juelich.de (H.F.)

**Keywords:** protein unfolding, cytochrome *c*′, small-angle neutron scattering, open-bundle structure

## Abstract

The dynamic structure changes, including the unfolding, dimerization, and transition from the compact to the open-bundle unfolding intermediate structure of Cyt *c*′, were detected by a small-angle neutron scattering experiment (SANS). The structure of Cyt *c*′ was changed into an unstructured random coil at pD = 1.7 (R_g_ = 25 Å for the Cyt *c*′ monomer). The four-α-helix bundle structure of Cyt *c*′ at neutral pH was transitioned to an open-bundle structure (at pD ~13), which is given by a numerical partial scattering function analysis as a joint-clubs model consisting of four clubs (α-helices) connected by short loops. The compactly folded structure of Cyt *c*′ (radius of gyration, R_g_ = 18 Å for the Cyt *c*′ dimer) at neutral or mildly alkaline pD transited to a remarkably larger open-bundle structure at pD ~13 (R_g_ = 25 Å for the Cyt *c*′ monomer). The open-bundle structure was also supported by ab initio modeling.

## 1. Introduction

Protein molecule is folded into a distinctive structure to express the unique role in biological systems. Anfinsen showed that most small proteins fold spontaneously into their specific functional structure [1,2]. Many protein-folding/unfolding experiments have been performed to date to reveal protein self-assembly mechanisms, and several general mechanisms have been proposed. The “energy landscape” model is a widely accepted hypothesis for describing protein folding pathways. This model proposes that the protein folding reaction explores exothermic conformational arrangements of the polypeptide along the potential energy surface, leading to the native protein structure. Many non-native protein structures are trapped in local minima on the energy surface. The structurally unidentifiable non-native state (molten globule) is presumed intermediate during the folding/unfolding reaction [3,4,5,6,7]. Numerous kinetic studies have probed the structures and properties of transient intermediate states [8,9] using far-UV CD, FT-IR, fluorescence, and NMR spectroscopy, as well as small-angle X-ray/neutron scattering and dynamic light scattering [7,10,11,12,13,14,15,16]. Englander et al. reported that the protein domains of cytochrome *c* and RNase are sequentially stabilized and folded in an orderly manner [17,18,19].

Together with experimental studies, molecular dynamics (MD) simulation is a useful technique to understand protein folding/unfolding mechanisms [20]. MD simulation of the unfolding of lysozyme showed that unfolding is triggered by the loss of hydrophobic contacts at the inter-domain surface of the enzyme [21]. Lindorff-Larsen et al. reported that protein folding first occurs locally, followed by stabilization of the local structural elements accompanied by intermolecular interactions, which further promote the folding process [22].

Structural characterization of the folding/unfolding intermediate species is key to understanding the principles underlying protein structure formation. The application of small-angle X-ray (SAXS) and neutron scattering (SANS) techniques to protein molecules provides information on their flexibility, size, and structural morphology in solution [23,24]. The unfolding intermediate of lysozyme was previously characterized using SAXS [16]. The unfolding of lysozyme is triggered by extension of the β-domain, which preserves the folding of the α-domain. Synchrotron radiation SAXS rapidly provides data compared to SANS experiments. The combination of flow-cell experiments [25] and size exclusion chromatography [26] with SAXS reduces radiation damage to and aggregate formation by protein molecules.

SANS experiments are very useful for determining the solution structure of a protein without radiation damage. The energy of the neutron beam used for SANS experiments is on the order of meV, which is almost identical to the energy of the infrared region. Very recent progress in the ab initio structure analysis of small angle scattering data has made possible “low-resolution structure models” that represent the overall shape of the solution protein structure [11,27,28].

Cytochrome *c*′ (Cyt *c*′) is a member of the *c*-type cytochrome family. Cyt *c*′ comprises a four-α-helix bundle structure, in contrast to cytochrome *c*, with a heme prosthetic group embedded in the C-terminal region (Figure 1). Cyt *c*′ undergoes a unique pH-dependent spin state transition. The spin state between pH 3 and 7 is likely a quantum mechanical admixture of an intermediate-spin (IS: S = 3/2) and a high-spin (HS) state. The spin state transition into the HS state at alkaline conditions (8 < pH < 12) is triggered by breakage of the inter-helix hydrogen-bonding linkage between helix C and helix D [29,30,31,32]. The structure of Cyt *c*′ at pH ~13 was suggested to be a six-coordinated low-spin (LS) state with imidazole (His120)/OH^−^ or two OH^−^ as the axial ligands, resulting in an “open-bundle” structure at pH ~13, as reported previously [33].

With these points in mind, we determined the SANS solution structures of Cyt *c*′ under different pD conditions to better understand the mechanism underlying the alkaline structure transition of Cyt *c*′. This is the first report of the unfolding intermediate structure of Cyt *c*′ at pD ~13. Determining the structures of Cyt *c*′ in solution at pD 6.4, pD 9.6, and pD ~13 provides a general explanation of protein folding/unfolding mechanisms and the relevance of these mechanisms to the alkaline spin-state transition of Cyt *c*′.

## 2. Materials and Methods

### 2.1. Sample Preparation for the SANS Experiments

Cyt *c*′ was extracted from *Alcaligenes xylosoxidans* and purified by cation exchange chromatography (CM-Sephadex C-50, GE Healthcare, Tokyo, Japan) and size-exclusion chromatography (Hiload Superdex 200 pg, GE Healthcare), followed by re-crystallization according to a previous method [33]. The purity of Cyt *c*′ was checked by sodium dodecyl sulfate polyacrylamide gel electrophoresis (SDS-PAGE, Appendix A).

Samples for SANS experiments were prepared at pD 6.4 and 9.6 with 20 mM phosphate buffer. The sample solutions at pD 1.7 and ~13 were prepared by the addition of a small amount of DCl and NaOD to the phosphate buffer, respectively. The hydrogen atoms in potassium dihydrogen phosphate and dipotassium hydrogen phosphate were substituted with deuterium by the repeated evaporation/addition of D_2_O to the solution. The buffer solutions were prepared using D_2_O, KD_2_PO_4_, K_2_DPO_4_, DCl, and NaOD. A small amount of concentrated Cyt *c*′ D_2_O solution was added to the D_2_O buffer solution at the target pD (pD-jump method). The concentrations of Cyt *c*′ were 5.5 mg/mL at pD 1.7 and pD 6.4 and 5.3 mg/mL at pD 9.6 and pD ~13. The electronic absorption spectrum of Cyt *c*′ at each pD value was measured using a nanodrop-2000C spectrometer (Thermo Fisher, Dreieich, Germany) to check the electronic state of the heme in Cyt *c*′ (Appendix A). The pD values of the buffers and samples prepared in D_2_O were measured using a pH electrode and corrected to pD by adding 0.4, based on an earlier report [34]. The details of the sample are summarized in Appendix A.

### 2.2. SANS Measurements

The SANS experiments were performed at the KWS-1 beam line at the Jülich Centre for Neutron Science (JCNS) at the Research Neutron Source Heinz Maier-Leibnitz Zentrum (FRM II) in Garching, Germany [35] using 2 mm path length quartz cells. The wavelength (λ) of the incident neutron beam was 5 Å, selected using the velocity selector, with a spread ∆λ/λ = 10%.

The collimator and detector were configured symmetrically at 8 m and 4 m for the low-Q region, respectively, and the configuration for the high-Q region was 4 m for collimation and 1.5 m for detection. Background scattering from the buffer solutions and empty cell were obtained in the same configurations and used to subtract unwanted background scattering. A secondary calibration standard of Plexiglas was used to determine the scattering intensity on an absolute scale and to correct detector sensitivity. The parameters of SANS data collections were listed in Appendix A.

### 2.3. SANS Data Reduction and Analysis

Data reduction from raw SANS profile images to 1D SANS curves, including all the above-mentioned corrections and calibrations, was conducted using qtiKWS. Radial averaging converted the 2D data to 1D intensity profiles (I(*Q*)), with the scattering vector Q being calibrated by the wavelength and scattering geometry. The data obtained for each sample with different sample-to-detector distances were merged. The background was removed by subtraction of the scattering intensity from individual buffer solutions.

The radius of gyration (R_g_) of Cyt *c*′ at each pD condition was estimated from the linear low-*Q* region of the Guinier plot and the pair distribution function, *P*(*r*), using the PRIMUS program [36] in the ATSAS package. The gradual increase in the low Q intensity at pD ~13 could be due to a minor contribution of weak aggregation. The region corresponding to weak aggregation was not used for the Guinier and *P*(*r*) analyses. The data ranges for Guinier analysis and the calculation of *P*(*r*) were 0.001 Å^−2^ < Q^2^ < 0.002 Å^−2^ and 0.05 Å^−1^ < Q < 0.25 Å^−1^, respectively. R_g_ at pD 1.7 was estimated from fitting the SANS curve with the Debye function (*D*(*Q*)) corresponding to the scattering curve from a polymer in random walk conformation using a numerical model described previously [37]. The R_g_ values were also calculated from the crystal structures at pH 6.0 (PDB ID: 4WGZ) and 11.4 (PDB ID: 4WGY) using CRYSON [38]. Further detailed analyses (joint-club model, molecular weight estimation, bead modeling, and estimation of the aggregated particle fraction) were described in the Appendix B, Appendix C, Appendix D and Appendix E and Appendix A. The software employed in SANS data reduction, analysis, and interpretation are listed in Appendix A.

## 3. Results

SANS experiments with Cyt *c*′ were performed at pD 1.7, 6.4, 9.6, and ~13. Figure 2 shows plots of the SANS intensity of Cyt *c*′ as a function of neutron scattering momentum, *Q*. The SANS curves at pD 6.4 and 9.6 have almost identical profiles, with a shoulder band at 0.2 Å^−1^, which is absent at pD 1.7 and ~13. The scattering intensities at pD 1.7 and ~13 were lower than at pD 6.4 and 9.6, likely due to dissociation of the dimer to the monomer. Furthermore, the shoulder band at pD 6.4 and 9.6 observed at 0.2 Å^−1^ is reproduced in the simulated SANS curve for the dimer structure (Figure 2B,C), and thus is characteristic of the dimer structure of Cyt *c*′.

The experimentally obtained radius of gyration (R_g_) values of Cyt *c*′ at various pD were evaluated from the Guinier plot and *P*(*r*) function (Figure 3) and are summarized in Table 1 and Appendix A. The experimental and calculated R_g_ values from the crystal structures of Cyt *c*′ at pH 6.0 and 10.4 are summarized in Table 2. The experimental R_g_ values of Cyt *c*′ at pD 6.4 and 9.6 were 18–19 Å and 25–28 Å at pD 1.7. The larger R_g_ value at pD 1.7 is due to the expansion or oligomerization of Cyt *c*′. The SANS profile at pD 1.7 was evaluated using Debye function (Figure 2A) and was used to analyze disordered polymer structures [39,40]. The SANS profile at pD 1.7 was fit well using the Debye function and gave R_g_ = 25.7 Å, and it clearly showed that the structure of Cyt *c*′ transitioned to the unfolded random coil structure. The unfolded random coil structure of Cyt *c*′ at pD 1.7 indicated by the SANS data is consistent with the structure previously proposed based on CD and ESI-MS spectrometry data [33].

The crystallographic structure of Cyt *c*′ from *Alcaligenes xylosoxidans* was suggested to be a dimer, but the association state in solution was unknown [41]. The R_g_ values for the monomer and dimer structures of Cyt *c*′ were calculated using the crystal structures of Cyt *c*′ at pH 6.0 (PDB: 4WGZ) and pH 10.4 (PDB: 4WGY) and were 17–19 Å (Table 2). Thus, the calculated R_g_ values are in good agreement with the experimentally obtained R_g_ values (Table 1), clearly demonstrating that the quaternary structure of Cyt *c*′ is a dimer in solution at pD 6.4 and pD 9.6. The calculated SANS curves based on the crystal structure of dimeric Cyt *c*′ at pH 6.0 and 10.4 are shown in Figure 2 and are quite similar to the experimental SANS curves at pD 6.4 and 9.6, especially for the range of *Q* < 0.1 Å^−1^, which represents the global structure of Cyt *c*′. The similarity in the calculated and experimental SANS profiles strongly suggests that Cyt *c*′ is a dimer in solution.

Fetler et al. reported that differences between experimental and simulated SAXS curves in the high-Q region of aspartate transcarbamoylase are due to quaternary structural differences in the solution and crystal structures [42,43,44]. In the present study, the discrepancy between the experimental and simulated curves of Cyt *c*′ in the higher-Q region might similarly be due to quaternary structural differences in the dimeric structure in solution and in the crystal (Figure 2B,C). Kratky plot analysis of SANS profiles can distinguish flexible and rigid structures [45], with typical highly rigid folded states giving a bell-shaped curve in the low-*Q* region, a disordered flexible structure giving a plateau shape in the high-Q region, and lacking a bell-shaped curve in the low-*Q* region [46,47].

The Kratky plot of Cyt *c*′ at pD 1.7 shows an intensified plateau region in the high-*Q* region, without a bell-shaped profile in the low-Q region (Figure 4A), strongly suggesting that Cyt *c*′ becomes a very flexible random coil structure, in good agreement with the Debye function analysis and previous CD and ESI-MS experiments [33]. The Kratky plots of Cyt *c*′ at pD 6.4 and 9.6 (Figure 4B,C) are clearly bell-shaped profiles centered at Q = 0.1 Å^−1^ with a plateau shape in the higher-Q region (Q > 0.2 Å^−1^), indicating a flexible moiety in the protein structure. The Kratky plot at pD ~13 shows a weaker, broader bell-shaped pattern in the low-*Q* region, suggesting a different size and/or shape of Cyt *c*′ compared to the structure at lower pD values.

In SANS curves, the extrapolated intercept intensity (I(0)) is proportional to the molecular weight (MW) and the weight concentration (C) of the scattering molecules.

The *MW* was determined according to Equation (1):(1)$$MW=I\left(0\right)/C\frac{MW_{st}}{I{\left(0\right)}_{st}/C_{st}}$$
where *MW_st_, C_st_*, and *I*(0)*_st_* are the molecular weight, weight concentration, and extrapolated intensity of the standard sample, respectively [48]. The MW ratios at various pD values were calculated based on the *I*(0) and *C* values at pD 1.7 (*I*(0)*_st_*, *C_st_*) used as the standard sample. The *I*(0) values were obtained by extrapolation of the SANS I(*Q*) curves (Fourier transformation of *P*(*r*)) and the Guinier plot (Appendix A). The calculated MW ratios of Cyt *c*′ are summarized in Table 2. The MW ratios evaluated from Guinier analysis and *P*(*r*) at pD 6.4 and 9.6 were twice that of the ratio at pD 1.7. Thus, the MW ratios also strongly support the dimer structure of Cyt *c*′ at pD 6.4 and 9.6. The *I*(0) value estimated at pD ~13 is identical to the value at pD 1.7, further indicating that Cyt *c*′ exists as a monomeric structure at pD 13.

Cyt *c*′ has a four α-helix bundle structure. The structure at pH > 12 was suggested to be an “open-bundle” structure, which retains the α-helix structure [33]. The SANS curve at pD ~13 was numerically analyzed using a “joint-clubs model”, designed to describe the scattering patterns from four cylinder-shaped clubs connected by three short loops based on the zig-zag chain model described in earlier reports [49,50]. The “joint-clubs model” for analyzing the SANS curve at pD ~13 well reflected the “open-bundle” structure [33]. A schematic image of the joint-clubs model is shown in Figure 2D. Each α-helix of Cyt *c*′ was modeled as a rigid club. Fitting using this joint-clubs model indicated that the length (*L*) of each club is 31.5 ± 3.1 Å. The Cyt *c*′ structure consists of four α-helices: A(Ala3-Lys31), B(Asp37-Phe59), C(Ala76-Asp98), and D(Asp103-Arg124). The lengths of helices A, B, C, and D along each helix axis are 42.6, 35.4, 32.4, and 32.7 Å, respectively, as determined from the crystal structure (PDB ID: 4WGZ). The average length of the helices is 35.8 Å, which is in good agreement with the *L* value of 31.5 ± 3.1 Å calculated using the joint-clubs model. The average diameter of the clubs (*R*) was determined as 10.6 ± 0.9 Å by the fitting, in good agreement with an average diameter of the helices of ~10 Å in the crystal structure. Therefore, the joint-clubs model strongly supports the previously speculated “open-bundle” structure of Cyt *c*′ at pD ~13.

Ab initio analysis was conducted to clarify the low-resolution solution structure of Cyt *c*′. Figure 5 shows the ab initio bead models drawn at the same scaling factor level to compare the size of the protein at different pD conditions. The bead models at pD 6.4 (Figure 5A) and pD 9.6 (Figure 5B) gave essentially the same shape in the resolution of ab initio analysis. The bead model obtained at pD ~13 (Figure 5C) showed an extended structure. This elongation suggests two possibilities regarding the solution structure: the oligomerization of Cyt *c*′ or the formation of the “open-bundle” structure. The first possibility can be excluded immediately because the molecular weight ratio (Table 3) reflects the monomeric state, as described above.

We conducted docking simulation [51] of the helices to the elongated bead model structure at pD ~13 to clarify the suitability of the bead model to reflect an “open-bundle” structure of Cyt *c*′ (Figure 5C, bottom). The volumetric map generated from the bead model at pD ~13 was fitted with four ideal helices. Docking with helices A, B, C, and D did not converge in the given space of the volumetric map and thus we chose helix B as a representative helix structure because its length is similar to the average length of the four helices in Cyt *c*′. The loop structures connecting each helix were omitted. Figure 5C shows a possible arrangement of the four helices in the “open-bundle” form of Cyt *c*′ in the space of the volumetric map. This arrangement of the four helices is consistent with the joint-clubs model. The volumetric map at pD ~13 was also fitted to a monomer with the correlation coefficient 0.81 between the volumetric map and the volume calculated from the model (Appendix D “Ab initio bead modeling and analysis of bead model”). This volume is insufficient to accommodate the dimer structure. Therefore, the bead model for pD ~13 (Figure 5C) can reasonably be concluded to be the structure of the monomeric “open-bundle” structure with four α-helices, in contrast to the oligomerization of cytochrome *c* [52,53]. The orientation of helixes and conformation of linkers at pD ~13 were simulated by BUNCH program [54]. The BUNCH program gave the well fitting of the SANS curve by the theoretical scattering curve from the rigid four helices (helix A, B, C, D) connected by three linkers. Figure 6 and Appendix A show the BUNCH models given by the ten independent simulations. The inconsistency of the ten BUNCH structures suggests that the “open-bundle” structure may have structural flexibility by the linker parts. The bead model obtained by DENFERT would be an average structure for the most abundant “open-bundle” structure.

The oligomerization of cytochrome *c* is initiated by the displacement of the C-terminal helix domain in the monomer to the corresponding position of the other monomer in the dimeric structure [46]. The SANS experiment with Cyt *c*′ at pD ~13 clearly demonstrated the monomeric “open-bundle” structure, and the fraction of aggregated particles was estimated to be only ~0.2% from the low-Q intensity (Appendix E “Estimation of aggregated particle fraction at pD ~13”). Therefore, Cyt *c*′ does not undergo the intermolecular structural reassembly observed for cytochrome *c* [52].

## 4. Discussion

The pH-induced structural transition mechanisms of Cyt *c*′ were studied based on SANS solution structure determinations and were determined to be (i) random coil monomer at pD 1.7, (ii) folded dimer at pD 6.4, (iii) initial dimer dissociation at pD 9.6, and (iv) monomeric “open-bundle” structure at pD ~13. The comprehensive SANS analysis in the present study is consistent with previous spectroscopic studies (mass-spectrometry, CD/MCD spectroscopy) and precise crystal structure analyses. The relation between the pH-induced spin state and the structural transition of Cyt *c*′ was described previously [33,55]. The present SANS study showed that the unfolding intermediate at pD ~13 is key for elucidating the structural transition mechanism. The “open-bundle” structure at pD ~13 is the first folding/unfolding intermediate structure obtained for Cyt *c*′ and has not been determined by crystal structure analysis. The “open-bundle” structure of Cyt *c*′ as the unfolding/folding intermediate could provide insights into the initial or last step in the unfolding or folding process, respectively. Opening of the four α-helix bundle of Cyt *c*′ into the intermediate “open-bundle” structure could be induced by the disappearance of inter-helix hydrogen-bonds at alkaline pH, reported previously [33]. The Kratky plot at pD ~13 indicated the partial flexibility of the “open-bundle” structure. The previous ESI-MS study [33] also supports the rigidness of the “open-bundle” structure by the constant charge states. Thus, the ab initio analysis at pD ~13 was applied, and the obtained bead model was reasonable to explain the four helices of monomeric Cyt *c*′, whereas the linkers between each helix were not investigated in this study. Further investigations are required to know the structure and flexibility of the linkers by nuclear magnetic resonance spectroscopy, quasi-elastic neutron scattering experiment, and/or molecular dynamics simulation.

## Figures and Tables

**Figure 1 biomolecules-12-00095-f001:**
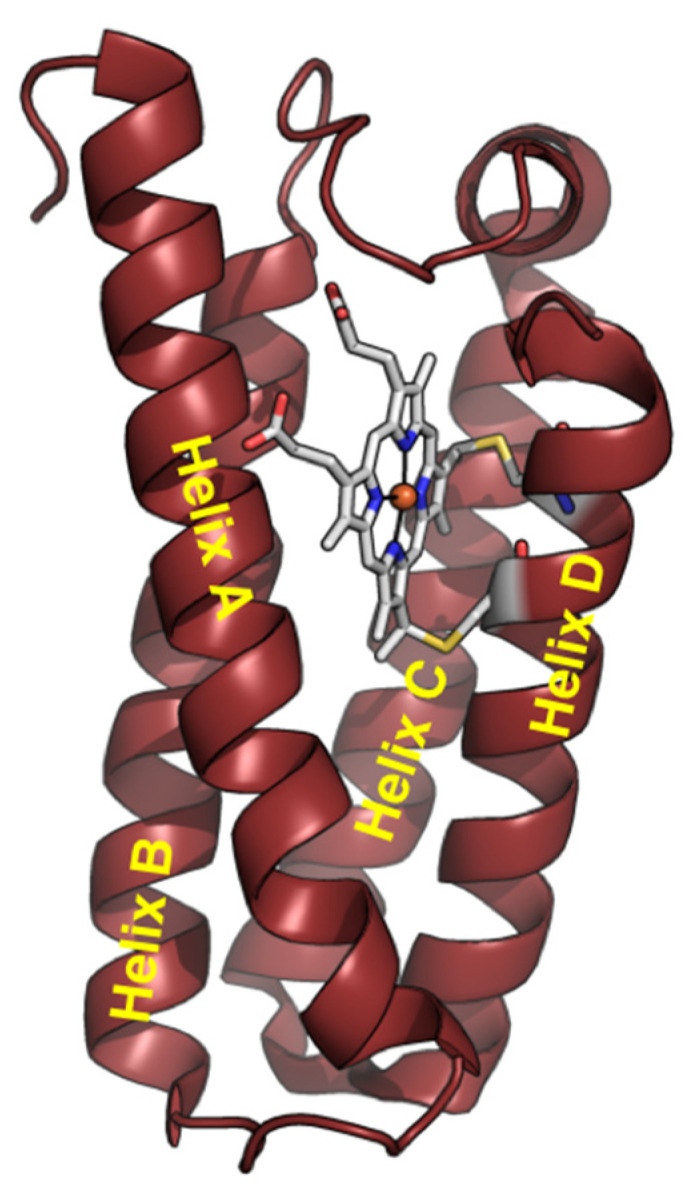
Structure of Cyt *c*′ monomer from *Alcaligenes xylosoxidans* NCIMB 11015 (PDB code: 4WGZ).

**Figure 2 biomolecules-12-00095-f002:**
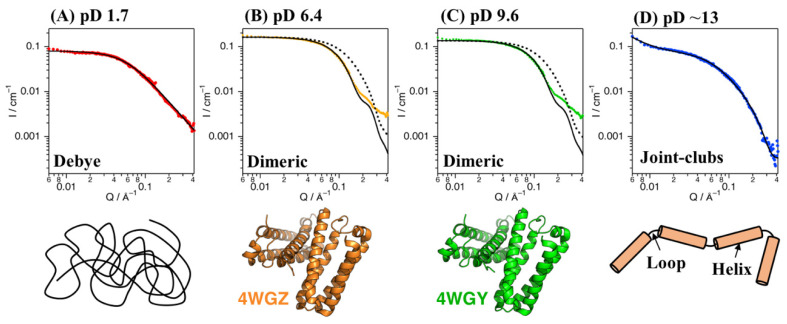
SANS data for Cyt c’ at pD 1.7 (**A**, red), 6.4 (**B**, orange), 9.6 (**C**, green), and ~13 (**D**, blue). The black solid lines represent the simulated SANS profiles for Cyt c’. Fitting of the Debye function and joint-clubs model are shown with the SANS curves at pD 1.7 and ~13, respectively. The simulated SANS curves obtained from the dimer (solid line) and monomer (broken line) crystal structures of Cyt c’ at pH 6.0 and 10.4 are shown in panels (**B**,**C**), respectively. Schematic representation of the random coil (**A**), folded dimer (**B**,**C**), and open-bundle (**D**) structures of Cyt c’ are provided under the respective SANS curves.

**Figure 3 biomolecules-12-00095-f003:**
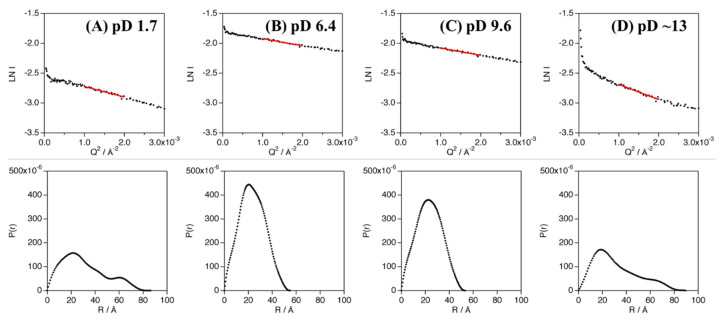
Guinier analysis (top panels) and *P*(*r*) function (bottom panels) of Cyt *c*′ at pD 1.7 (**A**), 6.4 (**B**), 9.6 (**C**), and ~13 (**D**). The linear regions of the Guinier plots used in the analyses are shown as red solid lines.

**Figure 4 biomolecules-12-00095-f004:**
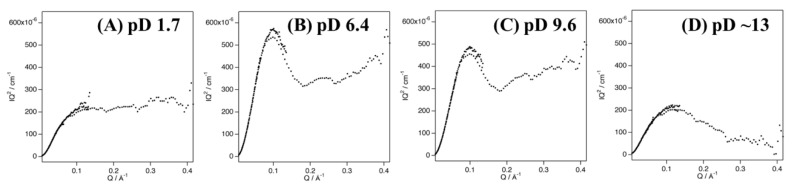
Kratky plots of the SANS curves of Cyt *c*′ at pD 1.7 (**A**), 6.4 (**B**), 9.6 (**C**), and ~13 (**D**).

**Figure 5 biomolecules-12-00095-f005:**
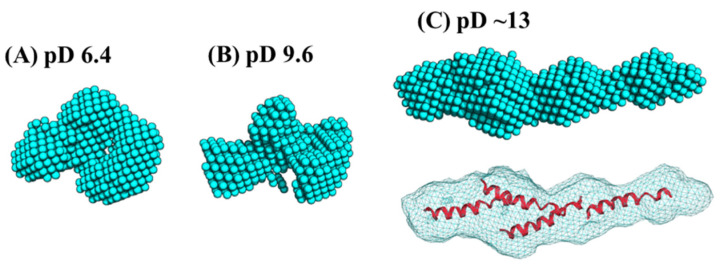
The ab initio bead models generated by the program DENFERT [27,28], and fitting of the crystal structure into each volumetric map (bottom panel) at pD 6.4 (**A**), 9.6 (**B**), and ~13 (**C**). The four B helices were docked in the volumetric map for (**C**) using the program Situs [51]. All models are drawn with the same scaling factor.

**Figure 6 biomolecules-12-00095-f006:**
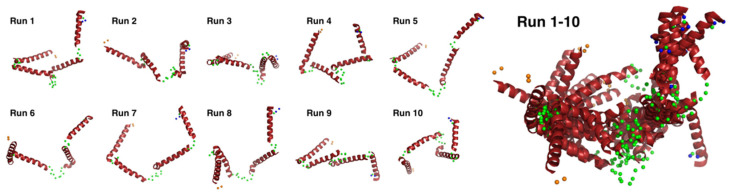
Ten individual BUNCH simulated structures and their superimposed structures. The BUNCH simulations were carried out with four rigid helixes (red), three linkers (green), N-terminal residues (orange), and C-terminal residues (blue).

**Table 1 biomolecules-12-00095-t001:** Estimated R_g_ values from experimental SANS curves.

Conditions	R_g_/Å		
	Guinier	P(r)	Debye
pD 1.7	23.02 ± 0.57	25.03 ± 0.19	25.65 ± 0.15
pD 6.4	18.93 ± 0.45	18.24 ± 0.08	n/a
pD 9.6	19.24 ± 0.49	18.10 ± 0.05	n/a
pD ~13	27.58 ± 0.59	25.54 ± 0.60	n/a

**Table 2 biomolecules-12-00095-t002:** Simulated R_g_ values calculated from the crystal structures at pH 6.0 (PDB 4WGZ) and 10.4 (PDB 4WGY).

Conditions		R_g_/Å	
		Guinier	P(r)
pH 6.0	Monomer	14.0	15.6
	Dimer	17.5	19.1
pH 10.4	Monomer	13.6	15.5
	Dimer	17.3	19.1

**Table 3 biomolecules-12-00095-t003:** Concentration (C) of Cyt *c*′, I(0) and estimated molecular weight ratio from experimental SANS curves at each pD condition.

Title 1	C (mg/mL)	Guinier		P(r)	
		I(0)	MW Ratio	I(0)	MW Ratio
pD 1.7	5.5	0.078	1.0	0.080	1.0
pD 6.4	5.5	0.165	2.1	0.161	2.0
pD 9.6	5.3	0.141	1.9	0.139	1.8
pD ~13	5.3	0.088	1.2	0.080	1.0

## Data Availability

The SANS data presented in this study are openly available in Small Angle Scattering Biological Data Bank, SASBDB IDs: SASDN52, SASDN62, SASDN72, and SASDN82.

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
