# Peer review of "Open-Bundle Structure as the Unfolding Intermediate of Cytochrome c′ Revealed by Small Angle Neutron Scattering"

_biomolecules, 2022, doi:10.3390/biom12010095_

Round 1
Reviewer 1 Report
Yamaguchi and col. Described the structure of an unfolding intermediate of cut c’ determined by SANS. I find this work really interesting and novel in terms of determining structural intermediates by SANS, but I feel that the authors do not unequivocally demonstrate in the manuscript that the structure of cut c' at pD 13 correspond to a random-coil structure. Here are some suggestions that might hopefully help:
.- Gel filtration, analytical ultracentrifugation, DLS, or any other technique that demonstrates the monodispersity of the protein preparation should be included.
.- What would be the results of analyzing the data at the 4 measured pDs with the “classical” ensemble optimization method (EOM)? How different is a dummy-bead model obtained from the experimental data at pD 1.7 from the one shown at pD 13?
.- In Figure 2, profiles at pD 1.7 and pD 13 should be shown overlapped, since they seem nearly identical at angles minor or equal to 0.1 inverse A. Similarly, at very low angle, scattering profiles significantly deviate from an ideal solution, which greatly difficult further analysis. Why is the profile obtained at pD 13 much noisier at high angle than the rest?
.- Docking/modelling of the helices to the dummy-bead ab initio model should be performed including the linkers between helices, which also contribute to the scattering. There are several software available for this task, i.e. BUNCH, GASBOR, etc.
Minor points:
.- Lines 77-80 make reference to AlphaFold but I do not understand what is the point here.
.- Line 180. Should be pD 9.6 instead of 9.4?
Author Response
Thank you very much for your careful reading of our manuscript. We appreciate your suggestion and question. Please see the attachment.
Reviewer 2 Report
In the manuscript "Open-Bundle Structure as the Unfolding Intermediate of Cytochrome c’ Revealed by Small Angle Neutron Scattering" by Yamaguchi et al. the authors investigate the alkaline transition of cytorchrome c' by small-angle neutron scattering (SANS). The experiments were performed at different pD so that different folding states of this protein could be determined.
The scattering curves were analyzed by a set of commonly used methods for such systems (e.g. Guinier analysis or Kratky plot) in order to gain extract information on the folding state and the e.g. radius of gyration. These values were compared with outcomes from simlations.
This experiments were well conducted and are scientifically sound. Moreover, the manuscipt ist well written, and easy to follow.
However, prior to publishing, the manuscript needs to be revised for orthografic errors (e.g. line 37, 162.. ) Also, the "Angström" symbol is broken in most cases.
Author Response

(The authors gave the same response as above.)

Reviewer 3 Report
This is a very interesting manuscript describing SANS experiments and modeling of cytochrome c' structural changes as function of pH. The results show significant changes in structure, including unfolding, dimerization, and transition from compact to open bundle. These changes are interpreted through advanced modeling. This work should be of high interest to a large scientific audience in protein biochemistry and biophysics.
My one comment is that the Abstract should be updated to better describe the interesting results.
Author Response

(The authors gave the same response as above.)

Reviewer 4 Report
This paper reports a small angle neutron scattering study of cytochrome c' at four pDs. The goal was to characterize the oligomeric state and degree of folding at the different pD values. SANS was used rather than SAXS to evade trouble with radiation damage with X-rays. The authors are economical in their presentation of the results. Their data suppprt their conclusions.
There are four problems with the paper that prevent acceptance in its present form. These problems should not take more than several days to address.
First, the paper does not report the accession numbers for deposition of the models and data in the Small Angle Scattering Biological Database https://www.sasbdb.org/.The small-angle scattering field expects that the scattering data and derived models are made available to the public upon publication.
Second, the presentation of critical parameter values does not follow closely the recommendations of the members of the International Union of Crystallography (IUCr) Small-Angle Scattering and Journals Commissions, the Worldwide Protein Data Bank (wwPDB) Small-Angle Scattering Validation Task Force and additional experts in the field. See https://doi.org/10.1107/S2059798317011597.
Third, the Discussion section contains the results of analyzing quantities and molecular models derived from the scattering profiles. These derived quantities are still results. The entire Discussion section needs to be moved to the Results section. The Conclusion section should be relabeled as the Discussion section. The Discussion section is where the new work is compared to prior art. A Conclusion section is not needed because each paragraph in the Results section ends with a concluding sentence.
Fourth, the paper also has some garbled sentences. For example, the first sentence of the last paragraph on Page 2 is very unclear.
Author Response

(The authors gave the same response as above.)
